# Invasive Stages within Alien Species and Hutchinson’s Duality: An Example Using Invasive Plants of the Family Fabaceae in Central Chile

**DOI:** 10.3390/plants11081063

**Published:** 2022-04-13

**Authors:** Ramiro O. Bustamante, Daniela Quiñones, Milen Duarte, Estefany Goncalves, Lohengrin A. Cavieres

**Affiliations:** 1Laboratorio de Ecología Geográfica, Facultad de Ciencias, Universidad de Chile, Santiago 775000, Chile; daniela.quinones@ug.uchile.cl (D.Q.); milenduartem@gmail.com (M.D.); estefigoncalves@gmail.com (E.G.); 2Instituto de Ecología y Biodiversidad (IEB), Santiago 775000, Chile; lcaviere@udec.cl; 3Departamento de Botánica, Facultad de Ciencias Naturales y Oceanográficas, Universidad de Concepción, Concepción 4091124, Chile

**Keywords:** biological invasions, species distribution models, Fabaceae, invasion risk, south-central Chile

## Abstract

To understand the factors that limit invasive expansion in alien species, it is critical to predict potential zones of colonization. Climatic niche can be an important way to predict the potential distribution of alien species. This correlation between niche and geographic distribution is called Hutchinson’s duality. A combination of global and regional niches allows four invasive stages to be identified: quasi-equilibrium, local adaptation, colonization and sink stage. We studied the invasive stages of six alien leguminous species either in the niche or the geographical space. In five of the six species, a higher proportion of populations were in the quasi-equilibrium stage. Notably, Acacia species had the highest proportion of populations in local adaptation. This picture changed dramatically when we projected the climatic niche in the geographic space: in all species the colonization stage had the highest proportional projected area, ranging from 50 to 90%. Our results are consistent with Hutchinson’s duality, which predicts that small areas in the niche space can be translated onto large areas of the geographic space. Although the colonization stage accounted for a low proportion of occurrences, in all species, the models predicted the largest areas for this stage. This study complements invasive stages, projecting them in geographic space.

## 1. Introduction

One notable feature that defines whether an alien species becomes invasive or not is its capacity to spread in the geographic space which in some cases encompasses large areas [1]. The causes of such expansion are still a matter of research, and understanding them provides important information for management and conservation initiatives [2]. Species distribution models (SDMs) constitute useful tools for examining potential geographic spread of alien species, linking niche attributes with environmental factors such as climate, soil type or vegetation type [3,4,5]. These models are based on two assumptions: (a) the niche of the invasive species is conserved, which means that good predictions are only possible if the environments in the invaded ranges are similar to those of the native range [6] and (b) after expansion, invasive species attain a biogeographic equilibrium, which means that they occupy every suitable habitat [7,8]. In many cases these assumptions are not met [9] because alien species do colonize regions with different environmental conditions than their native ranges, presumably due to niche shift [10,11] and the absence of dispersal limitations or natural enemies [12,13].

Invasive species offer an unprecedented opportunity to examine species niches from three perspectives: (a) global niche, constructed from all the environments and occurrences recorded for invasive species worldwide; (b) native niche, which considers the position of species in relation to the environment in its native range and (c) regional niche, which considers the position of species in the environment in some invaded region. Native and regional niches are the expression of the realized niche, while the global niche can be regarded as a proxy of the potential niche [5,14,15].

The native–regional niche contrast has been used in many studies to assess the potential of species to be successful in new ranges. However, given the complexities of invasion processes, this contrast is not enough to capture such complexity, as it requires that the assumptions of niche conservatism and biogeographic equilibrium are met [8], which does not always occur [14,16,17].

The use of the global–regional niche contrast [14,18] enables occurrence probabilities (P(O)) in a particular invaded range to be estimated using the potential niche of the species. The regional niche, in turn, is constructed using the presences recorded in a particular invaded range. By crossing the P(O) obtained from global and regional niches, we can display regional presences in a biplane to represent P(O) values projected from global and regional niche models. Thus, this plane expresses four invasive stages separated by a threshold of P(O) = 0.5: (a) if P(O) predicted from both global and regional niches are >0.50, then populations are in suitable habitats (quasi-equilibrium stage); (b) if the P(O) predicted from both global and regional niches are <0.50, then populations are inhabiting unsuitable habitats where populations do not persist (sink population stage); (c) if P(O) predicted from global niche models is >0.50 but the P(O) predicted from regional niche models are <0.50, then more suitable habitats remain to be occupied (colonization stage); (d) if P(O) predicted from the global niche are <0.50 but P(O) predicted from regional niche models is higher than 0.5, then the species is inhabiting novel environments not predicted from the global niche model (adaptation stage). Each of these stages can be considered as a biogeographical hypothesis to be tested by independent observations or by manipulative experiments [19].

Besides identifying the probability that populations are in a particular stage, populations can be also projected onto the geographical space. Using SDMs, it is possible to identify the areas in geographic space occupied by invasive stages. Here, we can link the proposal of [14] with Hutchinson’s duality [20]. In short, this duality indicates that a point in the multivariate niche (Hutchinsonian niche) corresponds to many points in geographic space (Grinnellian niche), while a point in geographic space corresponds to only one point in the multivariate niche [20]. The occurrences within invasive stages correspond to the Hutchinsonian niche [14], while the area of invasive stages projected on a map corresponds to the Grinnellian niche. As far as we are aware, no studies of plant invasion have been conducted that link these two conceptual frameworks. Following [20], we can expect no correspondence between the two niche concepts, i.e., that conditions that define a reduced set of niche conditions would be projected into larger areas of the geographic space.

In this study, we aim to complement the theoretical framework proposed by [14] using Hutchinson’s duality. Specifically, we categorized invasive stages for a set of exotic species using their occurrences in central Chile. Then, we projected invasive stages in the geographical space, categorizing exotic species using proportional area. Finally, we assessed the equivalence of both species’ configurations using ternary plots, which are triangular graphic representations of three variables that range between 0 and 1 and are of current use in different fields of ecology [21,22,23].

## 2. Results

### 2.1. SDMs from Global and Regional Climatic Niches

Global and regional SDMs showed a good matching using the AUC index. This index was > 0.8 in the great majority of models (Appendix A). The omission rate, i.e., the proportion of occurrences from Chile which are not predicted by the global niche (Appendix A), was lower than 11% in *A. dealbata*, *L. corniculatus*, *T. monpessulana*, *U. europaeus*, in *C. striatus* it was 16.1%; the highest omission rate was 25.68% in *A.*
*melanoxylon*.

The bioclimatic variables that showed the highest correlation differed among models and species (Appendix A). For global models the variable that had the highest correlation for *Acacia dealbata* Link, *Teline monspessulana* (L.) K. Koch and *Acacia melanoxylon* R. Br. was mean temperature of the coldest quarter. For *Ulex europaeus* Brot., *Cytisus striatus* (Hill) Rothm and *Lotus corniculatus* L. the highest correlation corresponded to the annual temperature range, minimum temperature of the coldest month and annual mean temperature, respectively. For regional models of *C. striatus*, *T. monspessulana*, *A. melanoxylon* and *Lotus corniculatus* the highest correlation occurred with annual precipitation, while for *A. dealbata* and *U. europaeus* it occurred with annual mean temperature and mean temperature of the coldest quarter, respectively (Appendix A). The predictions of each SDM for study species are given in the Appendix A. We found that the areal extent predicted by global niche models was higher than the areal extent predicted by regional models (Appendix A).

### 2.2. Invasive Stages

For all species, the proportion of occurrences by population in a quasi-equilibrium stage was greater relative to the other stages, with an average of 0.62, ranging from 0.42 (*A. melanoxylun*) to 0.81 (*C. striatus*). Colonization stages were on average 0.25, ranging from 0.19 (*C. striatus*) to 0.33 (*T. monpessulana*). For the local adaptation stage, we found an average of 0.13, ranging from 0 (*C. striatus*) to 0.34 (*A. melanoxylon*) (Table 1). For the sink stage, we found an average of 0.15, ranging from 0.06 (*A. dealbata*) to 0.23 (*C. striatus*) (Appendix A)

For all species, the proportion of predicted area in a colonization state was greater than for the other stages, with an average of 0.68, ranging from 0.52 (*A. dealbata*) to 0.91 (*Lotus corniculatus*). For the rest of the stages, the proportion of predicted area was highly variable, ranging from 0.008 to 0.20 for local adaptation and from 0.04 to 0.33 for quasi-equilibrium (Table 2).

### 2.3. Ternary Plots

In four of the six species studied, in the occurrence by population, the quasi-equilibrium stage was the dominant stage (each species with more than 60% of occurrences in that stage) (Figure 1). *Acacia dealbata* and *A. melanoxylon* showed intermediate values, expressing the highest proportions of populations (occurrences) in the local adaptation stage (between 30 and 40%). In contrast, for all species the proportional area projected for the colonization stage was the highest, with proportions varying between 90% and 100% (Figure 2). In other words, niche models predicted a large area for populations in the colonization stage but not colonized yet. The extent of this area can be observed in the maps in Figure 3.

## 3. Discussion

Global and regional climatic niche models can predict the potential occupation of a species in a given region. By crossing the P(O) of the two types of models we can dissect the invasive stages for each species in the niche space [14]. In this study, we proposed to complement this framework projecting the area of the invasive stage (Figure 3) using Hutchinson’s duality [20]. Note that the invasive stages proposed by [14] are regarded as biogeographic hypotheses to be tested by experiments, and using Hutchinson’s duality, we can predict the area of these stages and where they might occur. The Discussion is organized as follows: (a) exploring some hypotheses to explain the observed invasive stages for the alien species; (b) examining the discrepancies between invasive stages that emerged from the ternary plots; and (c) discussing the implications and utility of using Hutchinson’s duality in invasion ecology.

### 3.1. Invasive Stages and Hypothesis

From the framework proposed by [14], we found that the quasi-equilibrium stage was the most frequent stage across the six species we studied (more than 60% of populations for each species), suggesting that a large fraction of populations are in biogeographical equilibrium, i.e., there is a match between global and regional climatic niche. There are no species in which local adaptation is dominant; *Acacia dealbata* and *A. melanoxylon* tended to have the highest proportion of populations in the local adaptation stage (20–40% of populations), which means that they are partially colonizing novel environments not predicted by the global climatic niches. Independent studies have shown that *Acacia dealbata* has the potential to colonize climate zones not predicted by its native climatic niches [7]. Moreover, the *Acacia* genus is considered a climate generalist with the ability to colonize diverse environments due to plastic physiological and life-history responses or local adaptation, which enable it to colonize new environments [24].

The colonization stage was less frequent (40% of occurrences for all species). *T. monpessulana* and *U. europaeus* had a higher proportion of occurrences in the colonization stage, which means that there are still biotic limitations to colonizing suitable environments in central Chile. In the case of *T. monpessulana*, the availability of mycorrhizae and biotic pollination are limiting factors documented in other studies [25,26]. Moreover, the shade effects from remnant vegetation severely limit the regeneration of *Ulex europaeus* as well as flower and fruit production, with colonization being restricted to old fields and barren sites resulting from human disturbance [27]. Additionally, it is possible that *T. monpessulana* is an attractive resource for herbivores such as goats or insects, which prevent further expansion [28,29]. In the USA (invaded range), *T. monpessulana* is heavily consumed by the specialist eriophyid mite *Aceria genistae* [30].

### 3.2. Hutchinson’s Duality

In order to graphically show Hutchinson’s duality [20], we summarized our results in two ternary plots, one representing the occurrences within the niche space (Figure 1) and the other, the proportional areas predicted in the geographic space (Figure 2). There was a clear discordance: while according to Figure 1, we found five of six species located within the quasi-equilibrium stage, in Figure 2, the highest proportional area predicted for all species was for the colonization stage. Our results agree entirely with Hutchinson’s duality; although the colonization stage does not account for more than 40% of occurrences of the total species studied, the models predicted the largest areas, representing between 50 and 100% of the potential distribution of species (Table 1).

The fact that climatic niche models predicted large areas for the colonization stage opens the question about the causal factors for such a pattern and whether these areas will be colonized in the near future. Given that regional climate models are an expression of realized climatic niche, the factors that could prevent further colonization in areas well predicted by the global niche model are dispersal limitation [31] and/or negative biotic interactions such as interspecific competition or predation [32]. However, we cannot discount a bias in the distribution models due to low sampling effort for most alien species in central Chile [33], which could modify our prediction if the number of presences increases in the future [34]. Clearly, a well-designed research program dealing with the distribution of alien species in central Chile would surely improve the predictive value of the models.

Distribution models derived from climatic niche indeed are a coarse representation of species distributions because there are other factors that can also explain their distribution, however, these factors become important at lower spatial scales; there is a consensus in the literature that at a biogeographical scale (the scale of this study), climate is the main driver of species distribution.

In summary, our study confirms the fact that invasive processes are a complex phenomenon. Within only a single species it is possible to detect populations experiencing different invasive stages. Interestingly, for these six species we detected quasi-equilibrium in spite of other stages also being present. Hutchinson’s duality enables the invasive space to be projected in the geographic space, identifying the locations where processes such as niche conservatism to local adaptation are occurring within the geographic distribution. This information, besides providing a better understanding of invasive processes in the space, will help us to focus on the populations of alien species that deserve more attention for management, control or eradication. Finally, the use of climatic niche as a predictor of exotic species distribution will help to give us a reference point for future distribution shift due to climatic change.

## 4. Materials and Methods

### 4.1. Study Area

Chile has more than 700 alien plant species [33]. Among them, leguminous species are important in diversity and because they are recognized as very invasive and exert a significant impact on biodiversity, outcompeting native plants and modifying fire regimes [35,36,37]. In this study, we focused on six alien leguminous species to examine their invasive stages using the proposal of [14] and Hutchinson’s duality [20]. The species are *Acacia dealbata*, *Acacia melanoxilum*, *Cytissus striatus*, *Lotus corniculatum*, *Teline monpessulana* and *Ulex europaeus*. We selected these species because they are frequent and invasive from latitudes 30° to 43° S [33].

Regional presence data were recorded from the National Museum of Natural History herbarium at the University of Concepción. To complement this data, we also conducted vegetation sampling from 30° to 43° S in Chile (Figure 1) along two long geographical transects approximately 11 km apart, one located along the coast and the other in the central valley. Every 10 km we established plots (2 × 50 m) located along the verge of the road. We selected secondary or tertiary roads, controlling for anthropogenic disturbance and because roads are the most obvious corridors for the spread of invasive species [38]. The plots were divided into 10 sub-plots (2 × 5 m each), recording the presence of each of the six study species. The total database for the regional models contained 748 occurrences (Appendix A).

Global species presences were obtained from the Global Biodiversity Information Facility (GBIF; http://www.gbif.org/; accessed on 15 October 2019). Data were filtered using the following criteria: (a) data were selected from 1950 onward; (b) occurrences had a geographic error of less than 1 km and were properly georeferenced; (c) data vouchers contained the name of the botanists responsible for identification of samples; (d) duplicate data were eliminated. We considered only data within a 1 km grid in such a way as to include just one point per grid. The total database for the global models contained 19,429 occurrences (Appendix A). Analyses and selection of data were conducted using R (version 3.6.2) [39].

Climate variables at a 2.5 arc minutes spatial resolution were obtained from WorldClim (https://www.worldclim.org/; accessed on 20 August 2021). This database includes 19 climate variables obtained from precipitation and temperature records [40]. To reduce among-variable collinearity we used Pearson correlation tests, using ENMTools version 1.44 [41]. When the correlation between two variables was >0.7, we chose the variable with more biological importance. After correlation analysis, we selected six variables: annual mean temperature (Bio1); maximum temperature of the warmest month (Bio5); maximum temperature of the coldest month (Bio6); annual temperature range (Bio7); mean temperature of coldest quarter (Bio11) and annual precipitation (Bio12).

### 4.2. Species Distribution Models (SDMs)

For the SDMs we used Maxent [42]. This method correlates presences (1 km^2^ resolution) with climate variables to obtain occurrence probabilities P(O) throughout a maximal entropy function; Maxent constructs the climatic niche as well as the potential distribution model depicted on a map. For global and regional niche models we used 75% of the presences to develop the training model and 25% of the presences for testing the models. For model testing we used the AUC test, selecting models whose AUC ≥ 0.8. For parameterization regulation we used β = 1 [43]. For the global and regional climate background, we chose 10,000 random points. For each model, we constructed 50 replicates, presenting the average model. We estimated the omission rate, i.e., the proportion of occurrences observed in Chile which are not predicted by the global niche. The threshold utilized to discern between suitable and unsuitable environments was 10th percentile training presence. We also calculated the potential area predicted by the global and the regional niche for each species (Appendix A).

### 4.3. Invasive Stages

Following the proposal of [14], we contrasted global and regional niches for each alien leguminous plant selected in this study. For each presence from the global and regional model we extracted the P(O). These probabilities were depicted in the bi-dimensional plane. We also used this bi-dimensional plane to project invasive stages on a map, using the threshold P(O) = 0.5 which defines each invasive stage. Then, we assessed the extent of the area of each stage using QGIS 3.10.0.

### 4.4. Ternary Plots

To compare the equivalence of invasive stage configurations from the niche (occurrences) with the geographic space (areas), we used ternary plots. In our case, we decided to include the three invasive stages that occur within the climatic niches (global or regional niche): quasi-equilibrium, colonization and local adaptation. We did not include the sink stage because it was considered outside the climatic niche. We constructed two ternary plots: (a) proportion of presences (the niche space) and (b) proportional area covered by each invasive stage (the geographic space).

## Figures and Tables

**Figure 1 plants-11-01063-f001:**
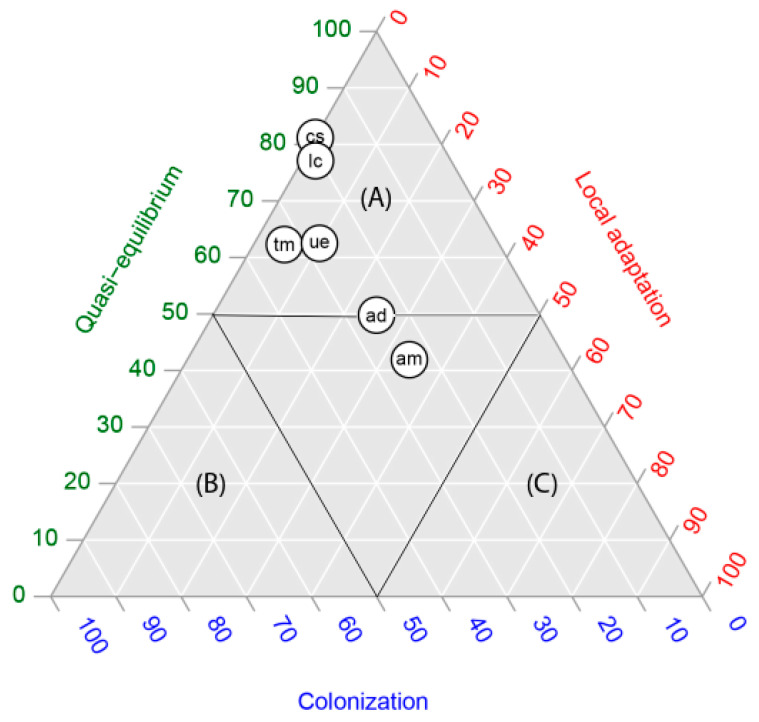
Ternary plots of the proportion of presences (the niche space) depicting the relative position of six alien leguminous species: *Acacia dealbata* (ad), *A. melanoxylon* (am), *Cytisus striatus* (cs), *Teline monpessulana* (tm), *Ulex europaeus* (ue) and *Lotus corniculatus* (lc), central Chile, in relation to quasi-equilibrium, local adaptation and colonization stages. Capital letters represent zones dominated by quasi-equilibrium (**A**), colonization (**B**) and local adaptation (**C**).

**Figure 2 plants-11-01063-f002:**
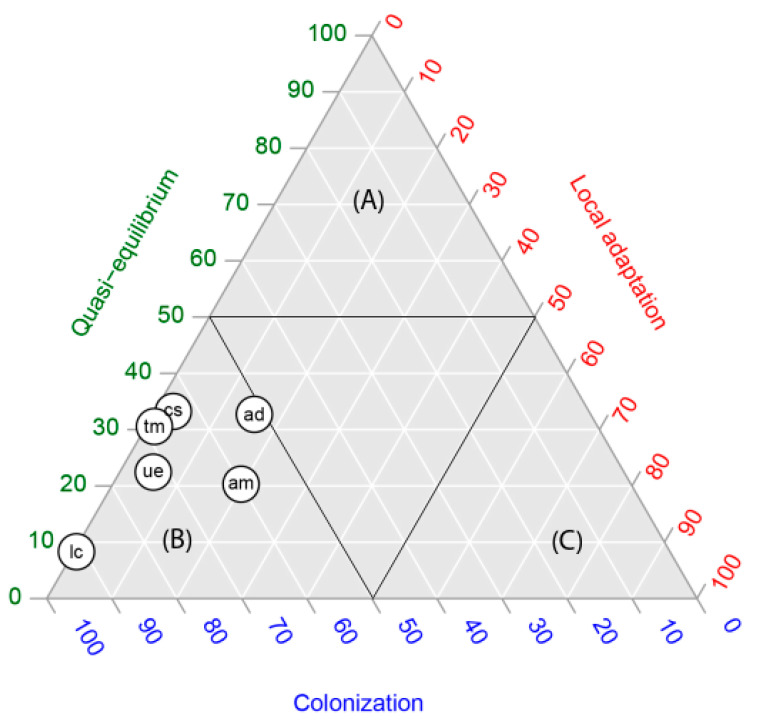
Ternary plot of the proportional area covered by each invasive stage (the geographic space), depicting the relative position of six alien leguminous species: *Acacia dealbata* (ad), *A melanoxylon* (am), *Cytisus striatus* (cs), *Teline monpessulana* (tm), *Ulex europaeus* (ue) and *Lotus corniculatus* (lc), central Chile, in relation to quasi-equilibrium, local adaptation and colonization stages. Capital letters represent zones dominated by quasi-equilibrium (**A**), colonization (**B**) and local adaptation (**C**).

**Figure 3 plants-11-01063-f003:**
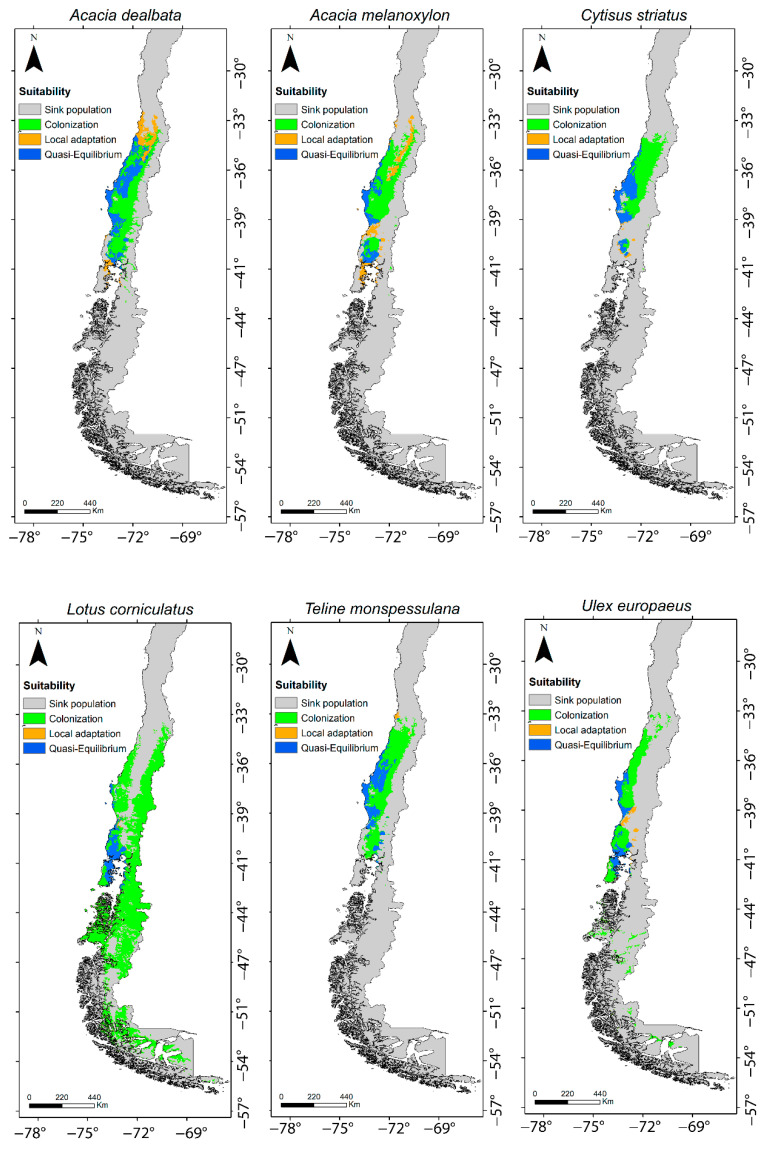
Areas of different invasive stages predicted by SDMs for the six study species. Gray represents predicted area for sink stage; green represents area for colonization stage; blue represents area for quasi-equilibrium stage; orange represents area for local adaptation stage.

**Table 1 plants-11-01063-t001:** Proportion of plant populations falling within the three invasive stages proposed by [14] for six alien leguminous species.

Species	Quasi-Equilibrium	Local Adaptation	Colonization	Total Occurrences (Sink Stage Excluded)
*Acacia dealbata* Link	0.50	0.25	0.25	243
*Cytisus striatus* (Hill) Rothm	0.81	0.00	0.19	48
*Teline monpessulana* (L.) K. Koch	0.62	0.05	0.33	106
*Acacia melanoxylon* R. Br.	0.42	0.34	0.24	129
*Ulex europaeus* Brot.	0.63	0.10	0.28	80
*Lotus corniculatus* L.	0.77	0.02	0.21	48
Average (Range)	0.62 (0.42−0.81)	0.13 (0.00–0.34)	0.25 (0.19–0.33)	109 (48–243)

**Table 2 plants-11-01063-t002:** Proportion of predicted area by regional and global models for the three invasive stages proposed by [14] for six alien leguminous species.

Species	Quasi-Equilibrium	Local Adaptation	Colonization	Total Area (km^2^)
*Acacia dealbata* Link	0.33	0.16	0.52	119.25
*Cytisus striatus* (Hill) Rothm	0.33	0.03	0.64	76.19
*Teline monpessulana* (L.) K. Koch	0.31	0.13	0.68	92.50
*Acacia melanoxylon* R. Br.	0.20	0.20	0.60	95.28
*Ulex europaeus* Brot.	0.23	0.05	0.72	78.80
*Lotus corniculatus* L.	0.08	0.004	0.91	233.30
Average (Range)	0.25 (0.08–0.33)	0.08 (0.004–0.20)	0.68 (0.52–0.91)	115.89 (76.19–233.30)

## Data Availability

Data used in the analysis are available in the Appendix A.

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
