# Peer review of "Invasive Stages within Alien Species and Hutchinson’s Duality: An Example Using Invasive Plants of the Family Fabaceae in Central Chile"

_plants, 2022, doi:10.3390/plants11081063_

Round 1

Reviewer 1 Report

The study presented is a good example of the methodological approaches to predicting distribution areas. Furthermore, it is good to use taxonomically more closely related species here. However, it would have been good to limit the study to woody plants and not to include a herbaceous species (Lotus). It would also have been good to include data on the invasion history globally and in Chile in a table, for example. Regarding the distribution maps based on the models, a comparison with the real distribution information from Chile and globally would be interesting. As important as the GBIF data are, they are unfortunately also incomplete. This is one of the reasons for the differences in the niche models (global/regional). And a very genaral problem. It would also be interesting to make a historical comparison. What influence does the duration of the presence of a species in an area have? The study presented is a good step forward. Due to the relatively good data situation in Chile (Concepcion herbarium), more extensive comparisons based on larger numbers of species would be very exciting. In this way, the following questions could be answered, such as. What influence do the life forms of the plants have? What is the significance of the geographical origin and, of course, the length of time since the species was first recorded in a region. A more detailed discussion is recommended.

Reviewer 2 Report

In this work, the authors studied the invasive stages (quasi-equilibrium, local adaptation, colonization and sink stage) of six alien leguminous species either in the niche or the geographical space using a Species Distribution Model approach. They performed their analysis in the framework of the Hutchinson’s duality.

Overall the manuscript is well written and presented, however they try to answer their hypothesis only using climatic variables (eg. topographic, edaphic, or habitat layers are not used. So the results in relation to the tested hypothesis (Hutchinson’s duality) are limited. This limit should be clearly addressed in the Discussion section

In addition, with regard to the SDM analysis additional information are needed (at least in the Supplementary material):

  • Model estimates are not reported
  • Model coefficients, variable importance are not reported
  • Model selection strategy is not reported
  • Performance statistics estimated on training data, on validation data and on test (truly independent) data are not reported

Line 112 - A. melanoxylon in italics

Line 228-229. Please insert the authorship for each species

Table S1 and S2. Table S1 and S2 are not reported in the Supplementary material

Round 2

Reviewer 2 Report

In this revised versione the authors provided information previously missed.

However, I think that in the discussion to address the limits of the study would be desirable. As pointed out in the previous revision, a key issue is that range maps are coarse representations of species distributions, and they are particularly prone to commission errors, where species are thought to be present in locations where they are actually absent (or viceversa).

Minor points:

P3L2. Check the sentence: a) species in italics, b) space between C. and striatus; remove exclamation point within brackets
